# Retreating Shorelines as an Emerging Threat to Adélie Penguins on Inexpressible Island

Xintong Chen [1], Jiquan Chen [2], Xiao Cheng [3,4], Lizhong Zhu [5], Bing Li [6] and Xianglan Li [1,4,*]

1 State Key Laboratory of Remote Sensing Science and College of Global Change and Earth System Science, Beijing Normal University, Beijing 100875, China; chenxt@mail.bnu.edu.cn
2 Department of Geography, Environment, and Spatial Science, Michigan State University, East Lansing, MI 48823, USA; jqchen@msu.edu
3 School of Geospatial Engineering and Science, Sun Yat-Sen University and Southern Marine Science and Engineering Guangdong Laboratory (Zhuhai), Zhuhai 519000, China; chengxiao9@mail.sysu.edu.cn
4 University Corporation for Polar Research, Beijing 100875, China
5 Heilongjiang Bureau of Surveying and Mapping Geographic Information, Harbin 150081, China; zlz@hlsm.mnr.gov.cn
6 Heilongjiang Geomatics Center of NASG, Harbin 150081, China; lib@hlsm.mnr.gov.cn
* Correspondence: xianglan_li@163.com

**Abstract:** Long-term observation of penguin abundance and distribution may warn of changes in the Antarctic marine ecosystem and provide support for penguin conservation. We conducted an unmanned aerial vehicle (UAV) survey of the Adélie penguin (*Pygoscelis adeliae*) colony on Inexpressible Island and obtained aerial images with a resolution of 0.07 m in 2018. We estimated penguin abundance and identified the spatial extent of the penguin colony. A total of 24,497 breeding pairs were found on Inexpressible Island within a colony area of 57,507 m$^2$. Based on historical images, the colony area expanded by 30,613 m$^2$ and abundance increased by 4063 pairs between 1983 and 2012. Between 2012 and 2018 penguin abundance further increased by 3314 pairs, although the colony area decreased by 1903 m$^2$. In general, Adélie penguins bred on Inexpressible Island at an elevation <20 m, and >55% of penguins had territories within 150 m of the shoreline. This suggests that penguins prefer to breed in areas with a low elevation and close to the shoreline. We observed a retreat of the shoreline on Inexpressible Island between 1983 and 2018, especially along the northern coast, which may have played a key role in the expansion of the penguin colony on the northern coast. In sum, it appears that retreating shorelines reshaped penguin distribution on the island and may be an emerging risk factor for penguins. These results highlight the importance of remote sensing techniques for monitoring changes in the Antarctic marine ecosystem and providing reliable data for Antarctic penguin conservation.

**Keywords:** UAV survey; penguin abundance; colony spatial extent; shoreline; elevation

## 1. Introduction

The Adélie penguin (*Pygoscelis adeliae*) is a circumpolar seabird that breeds on ice-free areas around Antarctica, usually close to open sea or polynya [1]. This species is very sensitive to physical changes in climate and landscape, and has been considered an eco-indicator of the Southern Ocean marine ecosystem [2]. Due largely to the rapid changes in the environment under the warming climate, penguin population dynamics in Antarctica have become a major focus in both scientific investigations and conservation [3–5]. Environmental variables, such as sea-ice concentration conditions, sea-surface temperature and food availability, are among the major drivers of penguin activities during their life cycle, including feeding, breeding, resting, etc. [3,6–10]. In addition, the dynamics and spatial distributions of penguin abundance have been viewed as warning signs of environmental changes and predictors in the conservation of other marine organisms in Antarctic ecosystems [11–13]. Consequently, effective and timely monitoring penguin populations is a key

element to the conservation of marine and land ecosystems in this fragile region under a changing climate [5,14].

Ice-free areas that are close to the open sea or polynya are the key habitats for penguin breeding [1], suggesting that both the amount and quality of these land areas directly affect the abundance and distributions of penguin colonies [15]. As in other regions of the Earth system, escalating climate change over the past decades has substantially reshaped Antarctic coastlines [16,17]. However, few attempts have been made to quantify how retreating shorelines have reshaped the abundance and distribution of Antarctic penguins. The Ross Sea is a high-quality chick-rearing habitat for the penguins in Antarctica [7]. Efforts have been made to map penguin colonies' distributions [18] and identify penguin abundance in the Ross Sea [3,19,20]. Lyver et al. [3] studied regional breeding population changes in the Ross Sea, and found that penguin abundance increased during 1981–2012. The authors attribute the increase to suitable environmental conditions and high food availability [3,11,13]. Several other initiatives quantify the changes to the shorelines and the surround landscapes [16,17], and these authors have reported physical environment changes along Antarctic coast which may affect penguins. On Inexpressible Island in the southernmost Adélie penguin colony in Victoria Land, over 17,000 pairs of Adélie penguins have been annually detected since 1983 [3,19–21].

A critical missing piece to our understanding about the changes in Adélie penguin abundance and distribution on Inexpressible Island has to do with shoreline dynamics. Our study objectives therefore are set: (1) to estimate penguin abundance in 2018 through identifying the "penguin" pixels and the spatial extent of the penguin colonies on Inexpressible Island using high resolution images; (2) to quantify the spatial variations in penguin distribution, colony extent, and shoreline dynamics in 1983, 2012 and 2018; and (3) to explore how the retreating shorelines have reshaped the abundance and distribution of Adélie penguins in this region.

Due to the obvious difficulties in conducting ground surveys of penguin distributions, remote sensing techniques are practical tools for detecting individual penguins across landscapes. Landsat satellite imagery with a resolution of 15 m has been successfully applied to map Adélie penguin colonies at a continental scale [18], and Quickbird-2 satellite imagery with a resolution of 0.6 m was used to detect changes in the locations of emperor penguin colonies in 2009–2013 [22]. In these studies, the relatively low resolution of these images prohibited direct estimates of numbers of individual penguins. However, higher resolution images recorded from aerial vehicles also can provide accurate identification of penguin colonies [23,24]. These spatial data can also accurately present the landform surrounding the colonies, allowing us to connect penguin distributions [25,26] with landscape structure (e.g., elevation and distance from the shorelines) [27,28]. More promisingly, unmanned aerial vehicle (UAV) surveys have been used to estimate penguin population size and to map penguin distribution [19,24,25]. Unfortunately, extreme weather conditions in Antarctica have made it difficult to perform aerial surveys at appropriate times (e.g., for breeding chronology) [26,29]. These practical challenges and associated high costs have prevented continuous monitoring of penguin abundance in the region. In this study, we were able to use relatively high resolution remote sensed imagery (0.07–0.15 m) from three periods (1983, 2012 and 2018) to assess long-term changes in penguin abundance and the habituating landscapes on Inexpressible Island to achieve our study objectives.

## 2. Materials and Methods

### 2.1. Study Site

We studied an Adélie penguin breeding colony on Inexpressible Island (74°54′S, 163°44′E)—a rocky island in the coastal zone of Terra Nova Bay in Victoria Land. The penguin colony covers approximately 300 m from east to west and 800 m from north to south. There is minimal human activity on Inexpressible Island, suggesting that the penguin colony is predominately affected by the environment and landscape settings. The first records of Adélie penguin abundance on Inexpressible Island were obtained for 1983,

when 17,120 breeding pairs were identified; this figure increased to 21,183 breeding pairs in 2012 breeding season [19].

### 2.2. Data Sources

The 34th Chinese Antarctica Research Expedition produced remote sensing images on 19 January 2018 from UAV flights. This expedition captured images of the coastal zone that covers the entire penguin colony at a resolution of 0.07 m (Figure 1). Adélie penguin abundance and the spatial extent of the penguin colony were quantified based on this database. We also constructed an orthophoto map covering penguin colony and a digital elevation model (DEM) for Inexpressible Island based on UAV images using PhotoScan Pro software (Agisoft LLC, St. Petersburg, Russia).

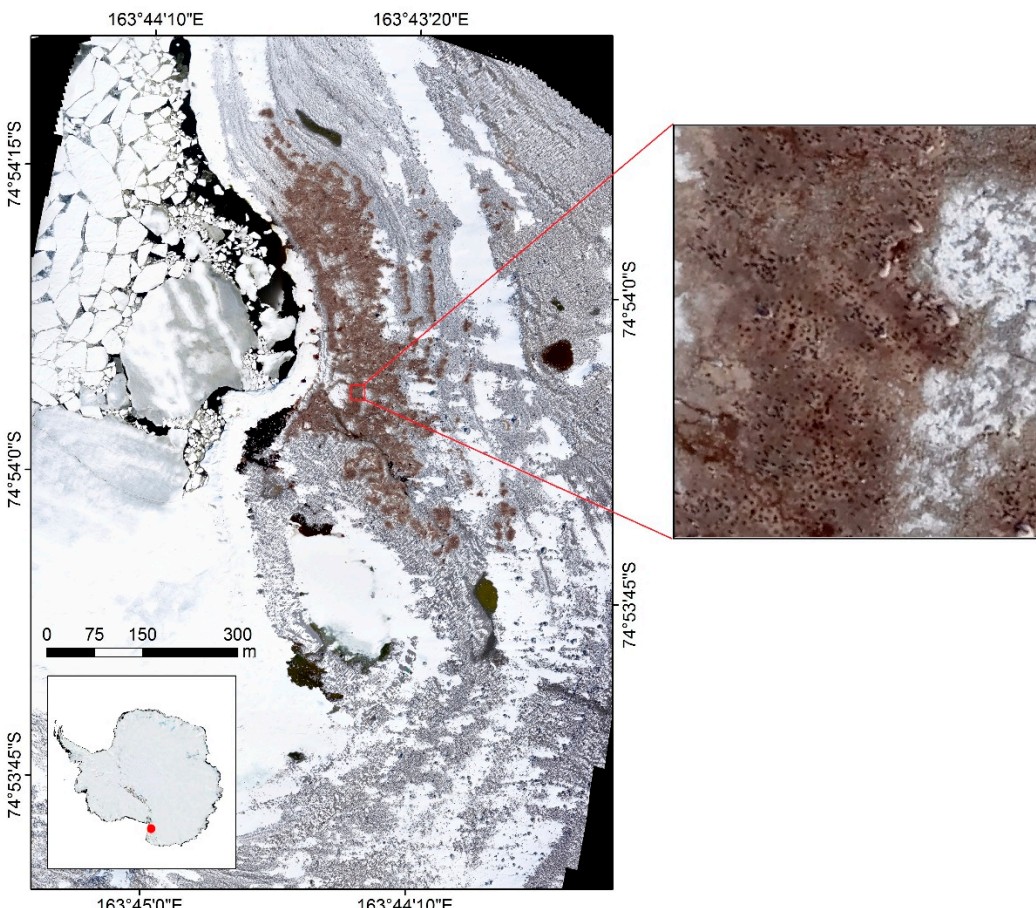

**Figure 1.** Orthophoto obtained during the UAV surveys on Inexpressible Island in the Ross Sea, Antarctica on 19 January 2018. Brown areas show the penguin colony. Black polygons (right panel) in the colony are penguin shadows.

We compared these images with historical high-resolution images and penguin population data to identify the distributions and sizes of the penguin colonies (Table 1). Historical aerial images of the Adélie penguin colony on Inexpressible Island taken in 1983 were obtained from the University of Minnesota (https://www.pgc.umn.edu/data/aerial/, accessed on 17 November 2021), and the UAV images of 2012 were obtained from the 29th Chinese Antarctic Research Expedition. Adélie penguin abundances on Inexpressible Island in 1983 and 2012 had been reported by He et al. [19]. Aerial images of 1983, 2012 and 2018 were orthorectified.

**Table 1.** Descriptions of the three remote sensing imagery datasets used for delineating shorelines, topography and penguin distributions on Inexpressible Island.

| Year | Type | Height (m) | Camera Type | Resolution (m) | Source |
|------|------|-----------|-------------|----------------|--------|
| 1983 | Aerial | 800 | WILDRC8 | 0.15 | University of Minnesota |
| 2012 | Aerial | 500 | Hasselblad H4D60 | 0.10 | The 29th CARE |
| 2018 | UAV | 350 | Hongpeng AP5100 | 0.07 | The 34th CARE |

Penguins travel 100–150 km from their colony during the breeding season to forage in the Ross Sea [30,31]. We therefore used a radius of 100 km around the colony site to consider the effects of sea-ice concentration (SIC), sea-surface temperature (SST), and chlorophyll concentration (CHL) on penguin abundance. We used satellite-derived data of SIC, SST and CHL. Monthly SIC data were obtained from the National Snow and Ice Data Center (https://nsidc.org/data/G02135/versions/3, accessed on 17 November 2021), with a resolution of 25 km × 25 km from 1980 to 2018. Monthly CHL data were obtained from MODIS-Aqua with a resolution of 4 km × 4 km from 2002 to 2018. Monthly SST data were received from the European Center for Medium-range Weather Forecasts (ECMWF) Reanalysis version 5 (ERA5) (https://www.ecmwf.int/en/forecasts/datasets/reanalysis-datasets/era5, accessed on 17 November 2021), with a resolution of $0.25° \times 0.25°$ from 1980 to 2018.

### 2.3. UAV Surveys

The Dajiang (DJI) Matrice 600 Pro UAV (SZ DJI Technology Co., Ltd., Shenzhen, China) was used with a Hongpeng AP5100 camera. The DJI Matrice 600 Pro carried an A3 Pro Flight Controller, with triple modular redundancy and diagnostic algorithms providing the centimeter-level accuracy. We designed the flight course and used the DJI mission-planning software to conduct the surveys. The field survey was performed during the 34th Chinese Antarctic Research Expedition on 19 January 2018. The flight path covered the entire penguin colony and the coastal zone. The total area covered by the UAV images was 3.48 km$^2$. The flight altitude was maintained at 350 m. The UAV course overlap and side overlap were >65%.

### 2.4. Shoreline Dynamics from 1983 to 2018

Based on initial visual interpretation, we extracted images of the shoreline from the aerial images of 1983, 2012 and 2018 within ArcGIS 10.4 to evaluate the changes in shoreline using the Digital Shoreline Analysis System (DSAS) [32]. We established a baseline that was generally parallel to the coastline and created 41 transects at 20 m intervals perpendicular to the baseline using DSAS. We then calculated the Net Shoreline Movement (NSM) as the distance between shorelines for each transect. The End-Point Rate (EPR) was calculated as the annual rate of shoreline change.

### 2.5. Data Processing

We extracted penguin-shadow pixels from the images and calculated the total number of penguin pixels within in the study area. The number of pixels corresponding to an individual penguin in the image was used to estimate penguin abundance in the study area. Object-based image analysis (OBIA) [19] was used to extract the penguin-shadow pixels from the UAV images to identify penguin presence through the Easy Interpretation software (Beijing Reavenue Technology Company, Beijing, China). Penguin-shadow pixels were then extracted from the UAV images and polygonised before tallying the total area of penguin shadow. According to the average height and chest width of Adélie penguin, and the solar elevation, the mean number of shadow areas of individual penguins was calculated following the penguin shadow analysis proposed by He et al. [19].

We used two quantitative measures to verify the accuracy of the classification: precision and recall. We also selected random areas within the penguin colony and performed visual interpretation of penguin abundance by calculating precision (Equation (1)) and

recall (Equation (2)) to determine the accuracy of the penguin pixels. Precision indicates the proportion of penguin pixels that was successfully extracted among all pixels; recall represents the proportion of extracted penguin pixels compared with all penguin pixels of the UAV images. These assessment metrics were calculated as:

$$Precision = \frac{TP}{TP + FP} \tag{1}$$

$$Recall = \frac{TP}{TP + FN} \tag{2}$$

where true positive (*TP*) represents actual penguin pixels that were successfully extracted; false positive (*FP*) values indicate pixels of other objects extracted by OBIA as penguin pixels; and false negative (*FN*) represents penguin pixels that were not extracted.

We examined the penguin distribution on Inexpressible Island by calculating penguin abundance and colony area by elevation classes of 0–10 m, 10–20 m, 20–30 m, 30–40 m, and >40 m for each year in 1983, 2012 and 2018. To further examine the changes in penguin distribution with distance from the shoreline, we divided the penguin colony into nine transects at distances from the shoreline of 0–50 m, 50–100 m, 100–150 m, 150–200 m, 200–250 m, 250–300 m, 300–350 m, 350–400 m, and >400 m using the shoreline in 1983. We first quantified the empirical influences of distance and elevation on penguin abundance and density for each year. A nested ANOVA (i.e., distance, elevation and year, with year nested within distance) was performed to tease apart the importance of these three independent variables. Within the ArcGIS, 100 points were randomly located on high-resolution images to create 30 m radius sampling plots for tallying penguin abundance and density. Prior to ANOVA, data were tested for normality and homogeneity using Shapiro's test and Levene's test with the shapiro.test function and leveneTest function in stats and car packages in R (version 4.0.2) [33,34]. Penguin abundance and colony area were also transformed to a normal distribution using the logarithmic transformation to meet the requirement of the nested ANOVA testing. All data used in analyses were obtained from the original remote sensing imagery.

## 3. Results

### 3.1. Shoreline Retreating

We found a rapid retreating of the shorelines from 1983 to 2018 (Table 2, Figure 2). The shoreline retreated 12.41 ± 2.50 m, or an annual average of 0.43 ± 0.09 m yr$^{-1}$ from 1983 to 2012. The maximum EPR of −2.12 m yr$^{-1}$ occurred on the north coastlines. The shoreline retreated further, by an average of 11.88 ± 0.88 m, from 2012 to 2018, or an annual average of 2.31 ± 0.17 m yr$^{-1}$, which is a 4.37-fold increase compared with the period of 1983–2012. The maximum NSM was −61.36 m from 1983 to 2012 and −27.63 m from 2012 to 2018. Along the coastline, the northern shoreline retreated faster than that in the south (Figure 2).

**Table 2.** The analysis of shoreline changes on Inexpressible Island in periods 1983–2012 and 2012–2018. NSM represents the distance between shorelines, EPR represents the annual rate of shoreline change.

| Shoreline Dynamics | 1983–2012 | 2012–2018 |
|---|---|---|
| Net Shoreline Movement (NSM, m) | | |
| Average | −12.41 | −11.88 |
| STD | 2.50 | 0.88 |
| Maximum | −61.36 | −27.63 |
| Minimum | −1.68 | −1.06 |
| End-Point Rate (EPR, m yr$^{-1}$) | | |
| Average | −0.43 | −2.31 |
| STD | 0.09 | 0.17 |
| Maximum | −2.12 | −5.38 |
| Minimum | −0.06 | −0.21 |

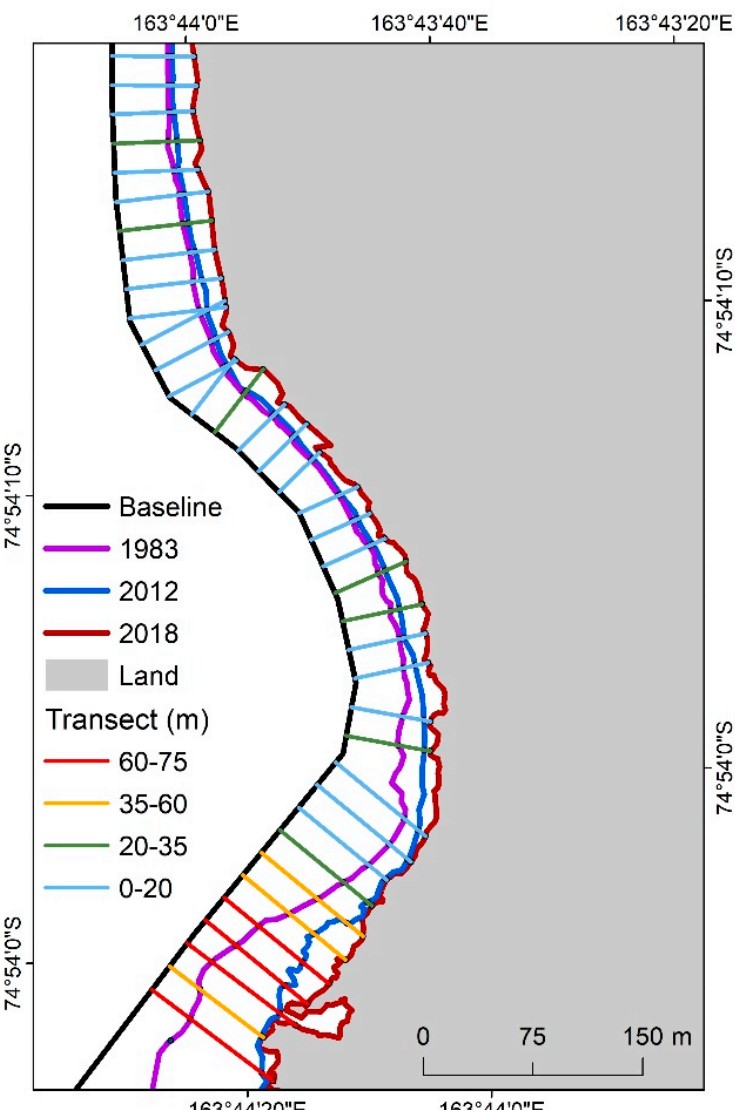

**Figure 2.** Dynamics of the shoreline on Inexpressible Island based on the Digital Shoreline Analysis System. The black line is the baseline. Cross-shore transects are each spaced by 20 m and shown in colors according to the distance between shorelines from 1983 to 2018. Shorelines in 1983, 2012 and 2018 are shown in purple, blue and red, respectively.

### 3.2. Adélie Penguin Abundance and Colony Area

The height and chest width of an Adélie penguin is estimated to be 46–61 cm and 15–25 cm [35]. The average height and chest width was set to be 53.5 cm and 20 cm, and the solar elevation was 33.77° when the UAV surveys were performed. The shadow analysis indicated that the shadow area of an individual penguin was 0.16 m$^2$. The land area with penguin-shadow polygons in 2018 was 3949 m$^2$. There were 24,497 breeding pairs of Adélie penguins on Inexpressible Island during the 2017–2018 breeding season within the study area along the coast (Figure 3).

To verify the accuracy of penguin identification, we randomly selected six sampling areas within the penguin colony, with size ranging 500–1500 m$^2$. TP, FP, and FN were then used to calculate the precision and recall based on visual interpretation and OBIA results (Table 3). The precision range was 0.79–0.89 and the recall fell within 0.88–0.98, which indicated high accuracy of penguin-pixel extraction.

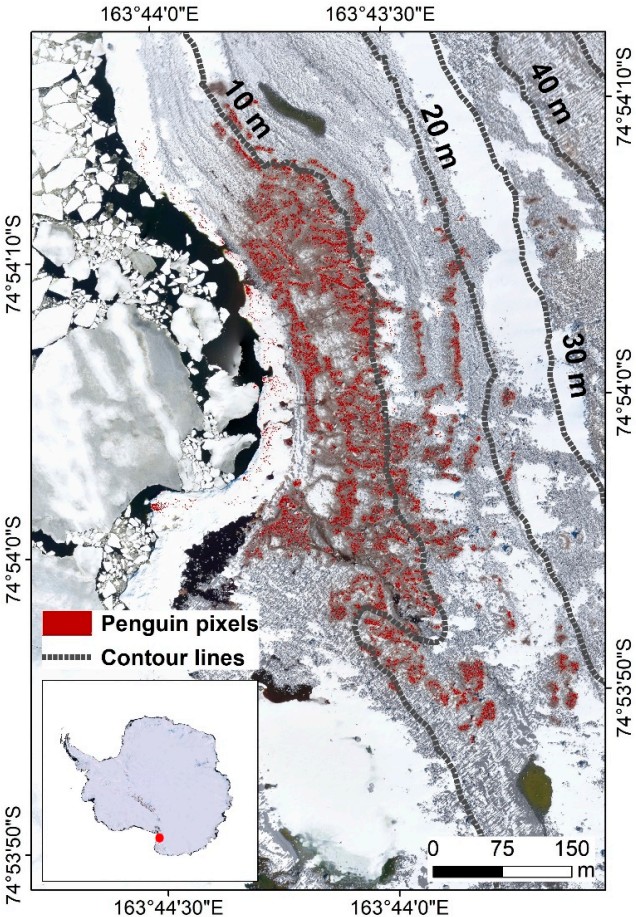

**Figure 3.** Extracted polygon mask of penguin pixels and contour lines from the UAV image and digital elevation model on Inexpressible Island in the Ross Sea, Antarctica on 19 January 2018.

**Table 3.** Accuracy assessment of penguin-pixel extraction results.

| Sample | TP | FP | FN | Precision | Recall |
|:------:|:----:|:---:|:---:|:---------:|:------:|
| 1 | 973 | 192 | 33 | 0.84 | 0.97 |
| 2 | 1080 | 157 | 39 | 0.87 | 0.96 |
| 3 | 576 | 73 | 75 | 0.89 | 0.88 |
| 4 | 483 | 122 | 23 | 0.80 | 0.95 |
| 5 | 956 | 252 | 23 | 0.79 | 0.98 |
| 6 | 730 | 142 | 26 | 0.84 | 0.97 |

The areas of penguin guano in the UAV images were delineated as the spatial extents of the penguin colonies in 1983, 2012 and 2018 (Figure 4). The penguin colonies were found in a strip along the coastal zone in all years. In 2018 the colony area covered 423 m from east to west and 776 m from south to north (total = 57,507 m$^2$). Penguin territories were generally within 250 m of the shore, except the northernmost part of the colony where their nests were near polynya. The colonized areas were 59,410 m$^2$ and 28,797 m$^2$ in 2012 and 1983, respectively. Overall, a significant expansion of colonized area was found from 1983 to 2012, especially in the northern region. It is worth noting that colonized areas between 2012 and 2018 overlapped by >90%.

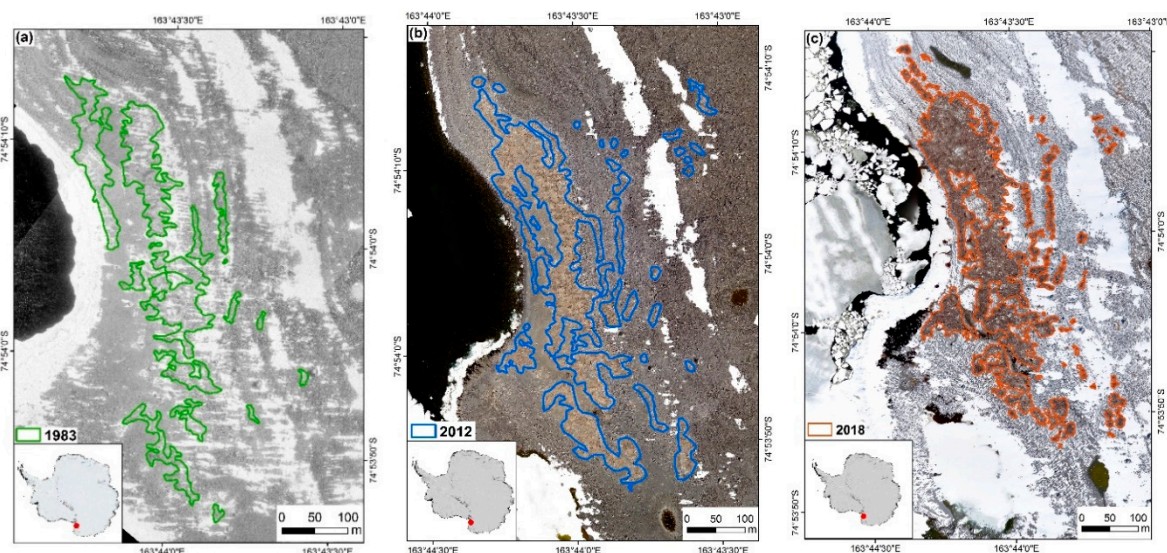

**Figure 4.** Spatial distributions of penguin colonies on Inexpressible Island in the Ross Sea, Antarctica during 1983–2018. Green, blue and red polygons show colony boundaries in 1983 (**a**), 2012 (**b**), and 2018 (**c**), respectively.

### 3.3. Changes in Penguin Distribution

According to records, the trend in penguin abundance on Inexpressible Island had a large annual fluctuation during 1983–2018 [19–21]. Their distribution with distance showed a right-skewed pattern from the shoreline for all three years (Figure 5a). It increased from 50 m, peaked at 100–150 m with 4585, 5128 and 6753 pairs in 1983, 2012 and 2018, respectively (Figure 5a), and declined at farther distances. We found 57%, 58% and 66% of the penguins were distributed within 150 m of the shoreline in 1983, 2012 and 2018, respectively. The colonized areas by penguins were 28,797 m$^2$ in 1983, 59,410 m$^2$ in 2012 and 57,507 m$^2$ in 2018. The change in colonized area with distance from the shoreline showed a similar trend to penguin abundance, although it peaked at 50–100 m (Figure 5b). For colony density, the average value was 0.59 pair m$^{-2}$ in 1983, 0.36 pair m$^{-2}$ in 2012 and 0.42 pair m$^{-2}$ in 2018. There appeared to be a decrease with distance, with the highest density found within 0–50 m of the shoreline (Figure 5c) with 0.71, 0.50 and 0.79 pair m$^{-2}$ for 1983, 2012 and 2018, respectively. Compared to that in 2012, penguins were distributed more evenly and with higher density in 2018. An apparent "outlier" was found at the distance of 250–300 m, where the density was very high in 1983 but low in 2012. For 2018, a small density peak was found at the distance of 350–400 m. Our analysis of variance of the penguin distribution with distance and year confirmed the significance of both independent variables (Table 4). In sum, the contribution of distance to the variance in penguin abundance and colony area was 60.61% and 55.57%, respectively, whereas the contribution of year to penguin density was higher (55.19%) than distance (25.73%).

We also examined the dependency of penguin distribution with elevation. The penguin colonized area decreased with increasing elevation, with 98%, 95% and 98% of penguins having territories at an elevation <20 m in 1983, 2012 and 2018, respectively (Figure 6). The colony expanded from 1983 to 2012 and 2018. Some territories (711 m$^2$) were found at elevation >30 m in 1983, but not in 2018. Some penguins were found at elevations of >30 m and >40 m in 2012, with abundance of 968 and 803 pairs, respectively. Interestingly, penguin density linearly increased with elevation in 1983, but showed no significant change with elevation in 2018. In 2012, when we found penguins at elevation ranges >30 m, there seemed to be an increasing trend within 0–40 m, but a low density at elevation >40 m. Our nested ANOVA indicates that both year and elevation are significant at $p < 0.01$ (Table 5). Overall, the contribution of elevation to the variance in penguin abundance, colony area and penguin density was 84.13%, 76.98% and 54.36%, respec-

tively, whereas the contribution of year to penguin density was relatively higher than to abundance and colony (34.30%, 13.46% and 18.84%, respectively).

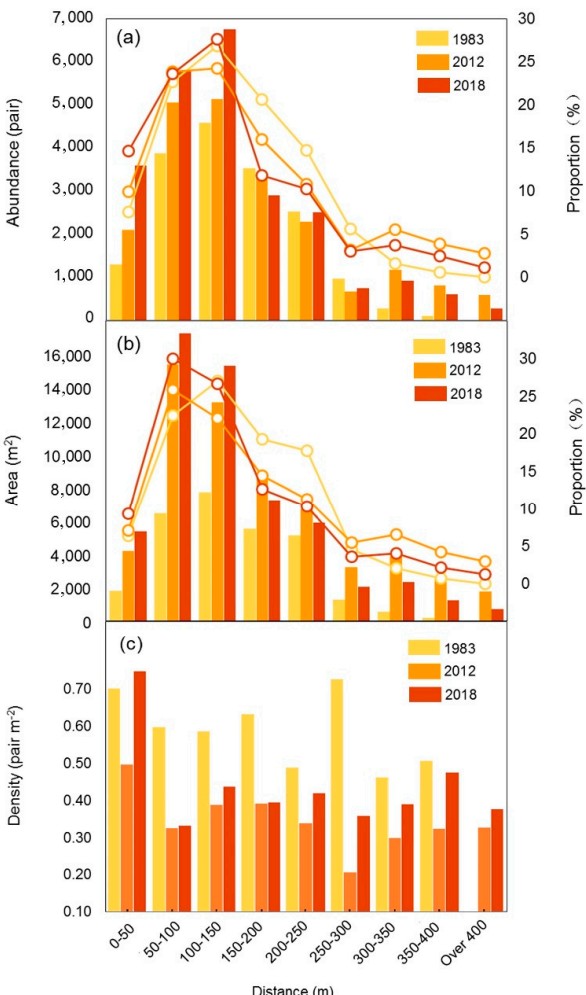

**Figure 5.** Penguin abundance (**a**), colony area (**b**) and density (**c**) at different distances from the shoreline. Histograms represent the total value and the lines are the relative proportion.

**Table 4.** Summary results of nested ANOVA analyses for determining the contribution of distance and year to penguin abundance, colony area and density between different distance ranges in study years. Year (Distance) represents year nested in distance in nested ANOVA analyses. * *p* < 0.01 according to nested ANOVA.

| Dependent Variable | Independent Variable | Sum of Squares | *df* | *F* | *p* | Contribution |
|---|---|---|---|---|---|---|
| Abundance | Distance | 27.40 | 8 | 66.92 | <0.001 * | 60.61% |
|  | Year (Distance) | 15.05 | 18 | 16.34 | <0.001 * | 33.29% |
|  | Residuals | 2.76 | 54 |  |  |  |
|  | Total | 45.21 | 80 |  |  |  |
| Area | Distance | 28.84 | 8 | 62.69 | <0.001 * | 55.57% |
|  | Year (Distance) | 19.95 | 18 | 19.27 | <0.001 * | 38.44% |
|  | Residuals | 3.11 | 54 |  |  |  |
|  | Total | 51.90 | 80 |  |  |  |
| Density | Distance | 0.62 | 8 | 9.20 | <0.001 * | 25.73% |
|  | Year (Distance) | 1.33 | 18 | 8.75 | <0.001 * | 55.19% |
|  | Residuals | 0.46 | 54 |  |  |  |
|  | Total | 2.41 | 80 |  |  |  |

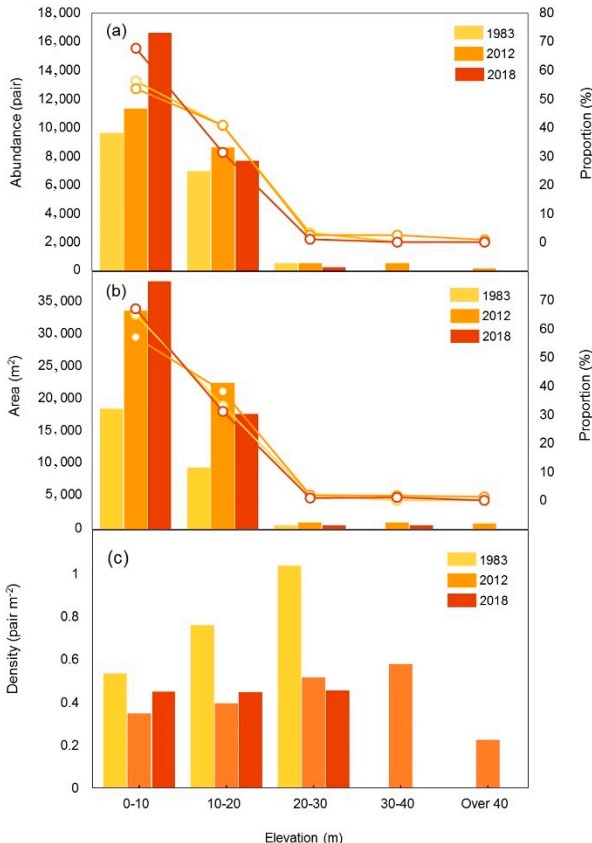

**Figure 6.** Changes in penguin abundance (**a**), colony area (**b**) and density (**c**) with elevation. Histograms represent the total value and the lines are the relative proportion.

**Table 5.** Summary results of nested ANOVA analyses for determining the contribution of elevation and year to penguin abundance, colony area and density between different elevation ranges in study years. Year (Elevation) represents year nested in elevation in nested ANOVA analyses. * $p < 0.01$ according to nested ANOVA.

| Dependent Variable | Independent Variable | Sum of Squares | df | F | p | Contribution |
|---|---|---|---|---|---|---|
| Abundance | Elevation | 87.13 | 4 | 261.64 | <0.001 * | 84.13% |
| | Year (Elevation) | 13.94 | 10 | 16.74 | <0.001 * | 13.46% |
| | Residuals | 2.50 | 30 | | | |
| | Total | 103.57 | 44 | | | |
| Area | Elevation | 81.28 | 4 | 138.21 | <0.001 * | 76.98% |
| | Year (Elevation) | 19.89 | 10 | 13.53 | <0.001 * | 18.84% |
| | Residuals | 4.41 | 30 | | | |
| | Total | 105.58 | 44 | | | |
| Density | Elevation | 1.87 | 4 | 35.84 | <0.001 * | 54.36% |
| | Year (Elevation) | 1.18 | 10 | 9.07 | <0.001 * | 34.30% |
| | Residuals | 0.39 | 30 | | | |
| | Total | 3.44 | 44 | | | |

## 4. Discussion

### 4.1. Factors Driving Adélie Penguin Abundance

Adélie penguins have existed intermittently in the Ross Sea for 45,000 years [36], across a latitudinal range of 1200 km [3]. The Adélie penguin abundance has increased over the past 30 years [3,6] and is projected to continue to increase in the Ross Sea [37]. Inexpressible Island is the southernmost Adélie penguin colony in Victoria Land, hosting >20,000 breeding pairs in recent years [1,3,6,19,20]. Our study found that Adélie penguin abundance on this island increased by 43.1% during 1983–2018. However, climate change

may alter physical and biological environmental conditions, which may affect Adélie penguin population dynamics [6,8]. Long-term monitoring and observation of penguin populations and colonies is thus urgently needed to improve our understanding of the effects of environmental changes on the Antarctic marine ecosystem [14].

Previous studies have indicated that the dynamics of penguin populations were jointly regulated by environmental variability and food resources [3,11,38]. However, changes in SIC and CHL on Inexpressible Island showed larger annual fluctuations, with no significant long-term trend in the breeding season (Figure 7). High SST may negatively influence penguin abundance by affecting penguin breeding performance [7,37,39]. The SST increased rapidly between 2005 and 2017—which may have a potential negative effect on penguin abundance [7,37,39], although it may be within the acceptable environment conditions for Adélie penguin survival. Glacial retreat and snow melt due to increasing temperatures, meanwhile, may have created more available habitat for Adélie penguins.

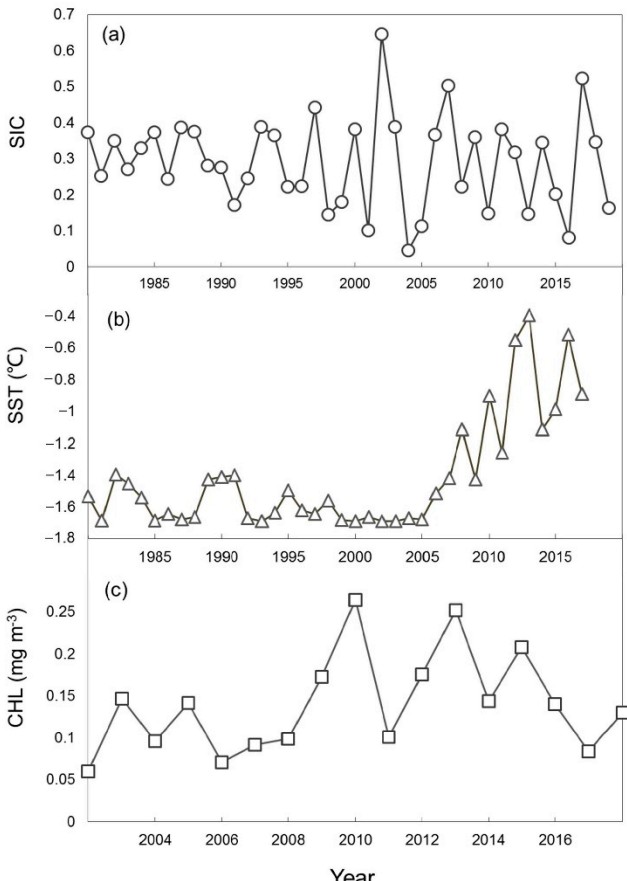

**Figure 7.** Changes in sea-ice concentration (SIC) (**a**), sea-surface temperature (SST) (**b**) and chlorophyll concentration (CHL) (**c**) in the Adélie penguin breeding season (December to February). SIC, SST and CHL were calculated for a 100 km radius around the colony site.

Recent studies have reported that variations in food resources might may the main factor driving penguin populations in the Ross Sea over the past 20 years [3,11]. Krill (*Euphausia* spp.) and silverfish (*Pleuragramma antarctica*) are the major prey species of Adélie penguins in the Ross Sea [11–13]. A commercial fishery for Antarctic toothfish (*Dissostichus mawsoni*), a competitor predator, may have benefited Adélie penguin population growth [3,11]. According to a Commission for the Conservation of Antarctic Marine Living Resources fishery report, toothfish fishing started in 1998, and 2000–3000 tons have been caught annually in the Ross Sea since 2004 [40]. Ainley et al. [41] and Lyver et al. [3] speculated that reduced trophic competition between penguins and toothfish increased the availability of food for Adélie penguins in the southern Ross Sea (Ross and Beaufort

Islands), resulting in an increase in the penguin population size in this region. This may also benefit penguins on Inexpressible Island and could be an important factor responsible for the increasing population.

### 4.2. Retreating Shorelines Affecting Adélie Penguin Colonies

The physical environment of the coastal lands may affect Adélie penguins' colony area on Inexpressible Island. Based on our UAV high-resolution images, we found considerable retreat of the shoreline on Inexpressible Island in 1983, 2012 and 2018, at an average rate of 0.71 m yr$^{-1}$ from 1983 to 2018, resulting in a maximum retreat of 70.95 m on the north coast (Table 2, Figure 2). At the same time, the area of the penguin colony increased by 1.06-fold from 1983 to 2012. The retreat of the shoreline might may been a key factor responsible for the new colony area on the north coast in 2012. We further hypothesize that a receding shoreline may affect the spatial extent of the colony, and the increased population size may also be an important factor driving the expansion of the colony during this period. However, the Adélie penguin abundance increased by 15.6% from 2012 to 2018, while the colony area remained the same, indicating that population size was not the only factor affecting the colony area. Based on nested ANOVA, we found that elevation and distance from the shoreline both significantly influenced the distribution of penguin abundance and colony area (Tables 4 and 5), which indicates that landform and access to food may influence the area of the penguin distribution. The Adélie penguin colony did not expand during 2012–2018, resulting in a higher colony density, and the colonies at elevations >40 m disappeared in 2018. Extreme weather conditions such as strong winds at high elevations may make these areas less suitable for penguin habitat. Penguins gather in areas within 100 m of the shoreline at an elevation of <20 m, indicating that they prefer territories close to the shore and at low elevations, likely for convenient foraging while avoiding the negative impacts of strong winds. Terrain and access to food thus have played a large role in restraining spatial colony expansion on Inexpressible Island.

### 4.3. Monitoring and Conservation of Adélie Penguins

Monitoring the distribution and abundance of Adélie penguins may provide basic data for Southern Ocean resource management [5,23]. Adélie penguins were classified as having a "vulnerable" conservation status according to the International Union for Conservation of Nature in 2012 [42], although this was changed to "least concern" in 2018 [43]. Adélie penguins have shown a trend toward increasing populations in the Ross Sea in past decades [3,6]. However, this increasing trend may not continue if there are continued changes in climate. Significant changes in the Antarctic coastline have occurred [16,17], with potential negative effects on the marine ecosystem. The collapsing ice shelf and retreating shorelines will reshape the physical environment of the Antarctic coast [17], and these retreating shorelines on Inexpressible Island will likely force the penguins to move inland to areas with higher elevations.

Penguin distribution and abundance may be affected by climate change in the future [37] and rapid decrease in penguin abundance has been projected under global warming [44]. The Ross Sea region has been considered the last refuge for penguins beyond 2100 [37]. However, retreating shorelines may impact some Adélie penguin colonies and have negative effects on long-term penguin abundance changes in the Ross Sea. The rising sea level may result in shoreline retreat on Inexpressible Island. We found that the annual retreat of the shoreline increased from 0.43 m yr$^{-1}$ to 2.31 m yr$^{-1}$ during the period of 1983–2012 to 2012–2018 (Table 2), and this may increase in the future, due to global warming. The shoreline will retreat 115.5 m in 50 years if the retreating rate we found in our study continues, which could destroy the main penguin breeding area on Inexpressible Island. Such habitat degradation has been considered a threat to penguins [45]. Retreating shorelines and the limitations of the terrain may force Adélie penguins to higher-elevation inland areas on Inexpressible Island, which could negatively affect penguin abundance in the future. However, in 2012, colony density was highest at an elevation >30 m and

<40 m (Figure 6), indicating that penguins do gather in areas with higher elevation. Colony density remained steady between each elevation range, and no penguins were found at an elevation >30 m in 2018, which suggests they prefer territories with low elevation on Inexpressible Island. Adélie penguin colonies are located in the coastal zone in Antarctica, and retreat of the coastline may thus present an emerging threat to their populations in some regions. There is thus a need to pay close attention to the effects of coastline retreat on penguin colonies and consider its potential effects on future penguin abundance changes.

## 5. Conclusions

We used high-resolution remote sensing images to estimate penguin abundance and colonized areas on Inexpressible Island in 2018 and made comparisons with historical records. We found 24,497 breeding pairs of Adélie penguins on the island in 2018. The total colony areas were 28,797 m$^2$ in 1983, 59,410 m$^2$ in 2012 and 57,507 m$^2$ in 2018. Colony density was highest in 1983 but decreased when the penguin colony area expanded in 2012. With increased abundance, penguins distributed more evenly and crowded more densely in 2018. Penguin abundance, landform and the retreating shoreline were found to be significant factors affecting penguin distribution. In sum, penguins preferred areas at low elevation and close to the shoreline (<150 m). With the shoreline retreat during 1983–2018, penguin populations were pushed further inland and to higher elevations, suggesting that the retreating shoreline due to the warming climate will be a serious threat to the Adélie penguin colony on Inexpressible Island in the future. In addition, application of aerial photography for monitoring long-term changes in penguin colonies is a practical approach for monitoring and conservation of this indicator species.

**Author Contributions:** X.C. (Xintong Chen) analyzed data and wrote the manuscript. J.C., X.L. and X.C. (Xiao Cheng) revised the manuscript. L.Z. and B.L. performed unmanned aerial vehicle surveys. All authors have read and agreed to the published version of the manuscript.

**Funding:** This study was supported by the National Key Research and Development Program of China (No. 2018YFC1406906).

**Institutional Review Board Statement:** This study was not involved with animal experiments, and all surveys were followed by Antarctic conservation measures.

**Informed Consent Statement:** Not applicable.

**Data Availability Statement:** The datasets used and/or analyzed during the current study are available from the corresponding author on reasonable request.

**Acknowledgments:** This study was supported by the National Key Research and Development Program of China (No. 2018YFC1406906). We thank the Polar Research Institute of China and Heilongjiang Bureau of Surveying and Mapping Geographic Information for their support for performing UAV surveys and field investigations on Inexpressible Island. We thank Kristine Blakeslee from the Department of Geography, Environment, and Spatial Science, Michigan State University for the excellent editing.

**Conflicts of Interest:** The authors declare that they have no competing interests.

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
