# Peer review of "Retreating Shorelines as an Emerging Threat to Adélie Penguins on Inexpressible Island"

_remotesensing, doi:10.3390/rs13224718_

Round 1

Reviewer 1 Report

Please see attached pdf

Reviewer 2 Report

In this work, Chen et al. conducted UAV survey to acquire high-resolution image to estimate and assess the abundance and areas of the Adélie penguin colony on the Inexpressible Island, Antarctic. The topic is interesting and the work provides some new insights on the spatial-temporal distribution of the penguin colony in regard with the changes in the environmental factors. The method adopted in this study using high-resolution aerial images also showed great advantages over the satellite data mentioned in the previous works. Given that there are increasing research attention on utilizing UAV for monitoring the coastal ecosystem, particularly in remote areas where access is difficult, this study is meaningful and should be able to draw more attention to the field. Lastly, the manuscript is well-organized and the language is highly coherent. I would recommend this work for publication in Remote Sensing.

Author Response

Thanks for the Reviewer 2 comments. We have already revised the manuscript for some minor grammar mistakes.

Reviewer 3 Report

This work expands on an earlier study by He et al. (2017) where Adélie penguin population on Inexpressible Island was estimated based on an Object Based Image Analysis (OBIA) method using high-resolution photos from 1983 and 2012. In this paper the same method is applied to estimate the Adélie penguin population (or abundance) for 2018 using a high-res image obtained by an Unmanned Aerial Vehicle (UAV). The spatial variations in penguin abundance and colony extent and density (distribution) between 1983, 2012 and 2018 is related to shoreline retreat and beach elevation using nested Anova analysis. The analysis indicates that Adélie penguins prefer to gather (i.e., breed) in areas within 150 m of the shoreline at an elevation below 20 m, indicating that they prefer territories close to the shore and at low elevations. The authors suggest that this is likely for convenient foraging while avoiding negative impacts of strong winds. The data also shows that the Adélie penguin population increased between 1983 and 2018 but the colony extent did not increase, resulting a higher colony density. The authors speculate that the increase in penguin abundance is attributed mostly to an increase in food availability as environmental indicators (changes in sea-ice concentration (SIC), sea surface temperature (SST) and chlorophyll concentration (CHL) of the sea water) do not show significant or trends positive to penguin abundance, though increasing temperatures may create more available habitat. The increasing rate of shoreline retreat found in the study suggests that the main penguin breeding area could be destroyed over the next 50 years, posing a considerable threat to the Adélie penguin population if climate change is not mitigated.

I enjoyed reading this manuscript. It is well written, the data and results are well presented and the introduction provides a sound background on the relevance of the performed study. I do have several concerns that I like to address. My first concern relates to the strong reliance on the work previously done by He et al. (2017) to estimate population abundance and distribution. I recognize that for this study a new image from 2018 is used, but it appears (to me at least) that the penguin breeding pairs estimated by He et al. (2017) are here again presented as newly obtained results (for example in Figure 4). It is very possible that the authors re-analyzed the images from 1893 and 2012 using the method of He et al. (2017) to obtain new results (or data), but I think that can be better clarified if that is the case. Otherwise, I would recommend to clearly cite He et al. (2017) when their data/results are presented.

Another concern that stood out to me is that two nested Anova analyses were performed to separately relate penguin abundance, colony area, and colony density to (1) distance from shoreline and (2) beach elevation. Would it not be more insightful if those two variables were analyzed together, to determine which factor most significantly influences penguin abundance and distribution? Also, distance from shoreline and beach elevation are very likely highly correlated, have the authors considered this when doing the analyses?

Lastly, I found the Discussion a bit lacking in scope. I appreciate that the authors try to explain the population changes to environmental drivers, but while the presented data (Fig.7) somewhat explain the increase (or rather refutes several explanations) of penguin abundance, it does not support their findings of why the penguins like to breed closer to the shorelines at lower elevations. It is suggested that strong winds at high elevations could make these areas less suitable for penguin habitat, but would that really have such a big impact over such relatively small differences in elevation? The authors do not support this statement with any data or analysis on wind regime in the area, so it cannot be determined whether this is true or not. Further, it seems logical that penguins like to be close to their food sources, but it seems that (according to Fig.5) that there is an optimum in penguin abundance at a certain distance from the shoreline. So, it would be insightful if this is investigated / discussed in more detail and is related to shoreline dynamics that might affect penguin populations. Also, it is stated that shoreline retreat negatively impacts penguin population, but now this is not supported by the findings as the population is increasing. It could be insightful if the authors can extrapolate their findings on distance and elevation in relation to penguin abundance to foreseen sea level rise. Can penguin breeding area move up shore with rising sea levels or is there a particular threshold (e.g., in elevation) where penguins will no longer be able to breed? Will it be a gradual process or suddenly after a particular tipping point? These discussions in my opinion can be helpful to better reflect on the main findings of the manuscript.

Smaller remarks

Line 113: 34358 breading pairs in 2018 are mentioned here, while the number 24497 breeding pairs for 2018 is presented later in the paper. Which number is correct, and where does the difference come from?

Line 120: A DEM is constructed from the UAV images. It would be useful to present this data, so more context is provided as where the population (for 2018) is located. Also, it is clear that an orthophoto is produced (Fig.1) but is not mentioned in the text that this is done and which software was used.

Line 186-191: These are results and should be moved to section 3.2.

Round 2

Reviewer 3 Report

Dear Editor,

The authors have adequately revised their manuscript according to the concerns raised and comments made. In the present form it is my opinion that the paper is suitable for publication.